# Coronal CT Attenuation Measurement for Osteoporosis Screening at the Proximal Femur: A Comparative Study with the Axial Approach

**DOI:** 10.3390/diagnostics15212794

**Published:** 2025-11-04

**Authors:** Kaifeng Ye, Junbo Qi, Jixing Fan, Yutian Luo, Yanlei Dong, Yuanyu Hu, Zhuo Chen, Yun Tian

**Affiliations:** 1Department of Orthopedics, Peking University Third Hospital, No. 49 North Garden Road, Haidian District, Beijing 100191, China; 2Engineering Research Center of Bone and Joint Precision Medicine, No. 49 North Garden Road, Haidian District, Beijing 100191, China; 3Beijing Key Laboratory of Spinal Disease Research, No. 49 North Garden Road, Haidian District, Beijing 100191, China

**Keywords:** osteoporosis screening, CT attenuation value, BMD, DXA, proximal femur

## Abstract

**Objectives**: There is currently a lack of research on coronal Computed Tomography (CT) attenuation measurements. This study aimed to evaluate a coronal CT attenuation measurement method for osteoporosis screening at the proximal femur and compare its performance with the conventional axial approach. **Methods**: A retrospective analysis was conducted on 708 patients who underwent both proximal femur CT and dual-energy X-ray absorptiometry (DXA) within a 6-month period between January 2013 and December 2023. The axial and coronal CT attenuation values for these patients were measured. The correlation, agreement, and efficacy in screening osteoporosis of the two methods were further evaluated and compared. **Results**: Both measurement methods demonstrated excellent inter-observer reliability. Axial measurements yielded slightly higher HU values than coronal measurements (mean difference: 3.93 Hounsfield unit (HU), *p* < 0.001), and the Bland–Altman analysis showed a Coefficient of Repeatability (CR) = 9.09 HU (95% CI: 8.63 to 9.61) between the two methods. A strong correlation was observed between the two measurements (Pearson’s coefficient (r) of 0.919 (95% CI: 0.907–0.930, *p* < 0.001)). Coronal measurements showed comparable correlation with femoral T-scores to axial measurements (0.766 (95% CI: 0.730–0.808) vs. 0.736 (95% CI: 0.697–0.771), *p* = 0.195). Coronal and axial measurements exhibit good predictive performance for osteoporosis diagnosis, with no statistically significant difference in AUC values between the two methods (0.850 vs. 0.864, *p* = 0.097). **Conclusions**: The coronal CT attenuation measurement method provides a reliable and complementary diagnostic tool for opportunistic osteoporosis screening, demonstrating strong correlation with axial measurements and comparable diagnostic accuracy.

## 1. Introduction

Osteoporosis, a common disease in elders, especially postmenopausal women, often increases the risk of fragility fracture and causes pain, disability, deformity, and even death [1,2]. Therefore, all diagnosed patients should receive comprehensive anti-osteoporosis treatment [3,4]. According to the recommendation of the World Health Organization (WHO), dual X-ray absorptiometry (DXA) is the standard for the diagnosis of osteoporosis [1,5]. Quantitative computed tomography (QCT) has also emerged as a reliable method for osteoporosis assessment [6,7]. However, both methods present limitations: DXA measurements are prone to inaccuracies due to spinal degenerative changes, while QCT requires specialized equipment and software, hindering its widespread adoption [8,9]. Consequently, many osteoporosis patients remain undiagnosed, leading to suboptimal detection rates.

In this context, opportunistic screening using conventional computed tomography (CT) scans offers a promising approach. Measuring CT attenuation values (in Hounsfield units) has gained recognition as a valuable method for osteoporosis screening [10,11]. It could be easily obtained from the existing routine CT examinations without additional radiation exposure and cost, and these advantages could promote its widespread application [12,13]. Previous studies have confirmed that the CT attenuation value of vertebrae can effectively reflect patients’ DXA results [14,15,16]. Furthermore, additional studies have demonstrated that proximal femoral CT attenuation values can serve as a reliable indicator for osteoporosis screening, with this parameter demonstrating a stronger correlation with the incidence of hip fractures [8,17,18]-one of the leading causes of mortality in elderly populations. Current studies have confirmed that multi-planar assessment of vertebral bodies enables a more comprehensive evaluation of osteoporosis status [9]. In contrast, the measurement of CT attenuation values at the proximal femur still relies on the axial plane in clinical practice, lacking supplementary planar measurements to assist in local bone quality assessment. The coronal plane at the proximal femur offers the advantage of covering a broader anatomical region, thereby supporting a more integrated assessment of overall bone mass. Although these benefits have driven research into coronal measurements of the proximal femur, their translation to clinical practice has been hindered by methodological challenges, primarily the complex ROI delineation or the dependency on 3D modeling techniques [19,20].

We introduce a coronal CT attenuation measurement method for the proximal femur. This study will (1) evaluate its correlation with femoral BMD T-scores and (2) compare its diagnostic accuracy for identifying osteoporosis against the established axial approach.

## 2. Materials and Methods

### 2.1. Ethical Consideration

This study was a retrospective analysis, and approval was obtained from the Medical Science Research Ethics Committee of our hospital (No. M2024260). Due to the retrospective nature of the study, informed consent was waived.

### 2.2. Study Population

We retrospectively analyzed patients who underwent DXA testing at our institution between January 2013 and December 2023. The inclusion criteria were: (1) age >18 years; (2) availability of both proximal femur CT and DXA scans of the same side, obtained within a 6-month interval. Exclusion criteria comprised: (1) absence of proximal femur CT images; (2) conditions precluding accurate CT measurement, including proximal femur fractures, tumors, metabolic bone disease, or prior hip surgery history.

### 2.3. Bone Mineral Density Evaluation

Bone mineral density (BMD) assessments were conducted for all patients using DXA technology (Discovery A series, Hologic Inc., Bedford, MA, USA). The scanning protocol included evaluation of both the lumbar spine (vertebrae L1-L4) and bilateral hip regions. For diagnostic classification, T-scores were calculated using reference data from the Third National Health and Nutrition Examination Survey (NHANES III) [21]. According to the WHO’s diagnostic criteria, patients were classified into three groups based on the lowest T-score from lumbar spine and femoral neck measurements: (1) osteoporosis (T-score ≤ −2.5), (2) osteopenia (−2.5 < T-score < −1.0), and (3) normal bone mineral density (T-score ≥ −1.0) [22]. Subsequent comparative analyses were performed across these diagnostic categories.

### 2.4. CT Attenuation Measurement

The picture archiving and communication system (PACS) was used to calculate the CT attenuation value from the existing routine three-dimensional reconstructive CT (Dual Source Computed Tomography DEFINITION, tube voltage 120 kV, a slice thickness of 5 mm, Siemens, Munich, Germany). According to a previously reported method for measuring CT attenuation values (HU) in the proximal femur [8], the axial CT attenuation values were obtained by assessing cancellous trabeculae within the femoral neck and intertrochanteric region. The axial measurement plane was defined as the second axial slice located 10 mm inferior to the reference slice, where the femoral neck cortical bone first appears (Figure 1). We positioned a rectangular ROI on the axial measurement plane to encompass the maximal cancellous bone area while excluding cortical bone, then recorded the mean CT attenuation value.

Coronal CT attenuation measurements were obtained according to the following standardized protocol (Figure 2):Planar selection: Identify the coronal slice demonstrating the maximal cross-sectional area of intertrochanteric cancellous bone, which typically coincides with the central mechanical axis of the proximal femur.ROI placement: Position a rectangular ROI to fully encompass this cancellous bone area.Cortical exclusion: Carefully adjust the ROI to exclude the cortical bone boundaries.Data recording: Record the mean CT attenuation value (in Hounsfield Units) provided by the imaging software.

The above measurements were performed in the bone windows. Two researchers measured the CT attenuation value independently without knowing the result of the DXA for the interrater reliability test. Both investigators were board-certified orthopedic surgeons with demonstrated proficiency in the standardized measurement protocol.

### 2.5. Statistical Analysis

For the comparison among the normal BMD, osteopenia, and osteoporosis patients, a one-way analysis of variance (ANOVA) and the least significant difference (LSD) post hoc test were used for continuous, normally distributed variables. The Kruskal–Wallis and Dunn post hoc tests were used for continuous, non-normally distributed variables. Then, statistical analyses were systematically conducted to evaluate measurement reliability and diagnostic performance. Inter-observer reliability was quantified using intraclass correlation coefficients (ICCs), with ICC values >0.75 denoting excellent reproducibility. Method agreement was further verified through Bland–Altman analysis. The correlation between measured parameters and reference standard values (ipsilateral hip T-scores from DXA) was examined using Pearson’s correlation. Diagnostic accuracy of coronal CT attenuation measurements for osteoporosis screening was assessed via ROC curve analysis, with determination of the optimal diagnostic cutoff value. All analyses were performed using IBM SPSS Statistics (v27.0), MedCalc (v20.0.4), and OriginPro 2022, with statistical significance defined as *p* < 0.05.

## 3. Results

A total of 708 eligible patients were included in the analysis. The baseline characteristics of the patient groups are presented in Table 1. In the overall study population, the mean age is 64.24 ± 11.29 years, and female patients accounted for 76.6% (543/708). The median (interquartile range, (IQR)) interval between CT and DXA examinations was 20 (3–83) days. Significant between-group differences were observed among the normal BMD, osteopenia, and osteoporosis groups regarding age, axial CT attenuation, coronal CT attenuation, and femoral neck T-scores (*p* < 0.05).

Two independent orthopedic surgeons evaluated the measurement consistency between the axial and coronal measurement methods. The intraclass correlation coefficients (ICCs) demonstrated excellent agreement, with values of 0.843 (95% CI: 0.810–0.871) for the axial method and 0.862 (95% CI: 0.833–0.887) for the coronal method, confirming strong intra-method reliability for both methods (Table 2). The mean (± standard deviation) trabecular bone attenuation values at the proximal femur were 70.67 ± 42.30 HU (axial) and 66.13 ± 44.83 HU (coronal). Median values were 67.20 HU (IQR: 40.80–97.00) for axial measurements and 62.05 HU (IQR: 35.50–93.90) for coronal measurements (Figure 3). Paired *t*-test analysis revealed significantly higher attenuation values in axial compared to coronal measurements, with a mean difference of 3.93 HU (95% CI: 2.58–5.27, *p* < 0.001). Additionally, a Bland–Altman analysis was conducted between the two methods, and the results showed a Coefficient of Repeatability (CR) = 9.09 (95% CI: 8.63 to 9.61) (Figure 4).

Following normality testing, correlation analysis was performed between the measurements obtained by the two methods. The analysis revealed a strong correlation with a Pearson’s coefficient (r) of 0.919 (95% CI: 0.907–0.930, *p* < 0.001), indicating excellent agreement between the two measurement approaches (Figure 5a). Furthermore, we examined the correlation between measurements obtained by both methods and T-scores. The correlation coefficients between CT attenuation values (axial/coronal) and T-scores were 0.766 (95% CI: 0.730–0.808) and 0.736 (95% CI: 0.697–0.771), respectively (Figure 5b,c). No significant difference was observed between these correlation coefficients (*p* = 0.195).

For osteoporosis identification, both measurement methods demonstrated good diagnostic performance. The axial measurements yielded an area under the curve (AUC) of 0.864 (95% CI: 0.834–0.891) for predicting osteoporosis, while the coronal measurements achieved an AUC of 0.850 (95% CI: 0.803–0.878) (Figure 6). DeLong’s test indicated no significant difference in AUC values between the two methods (*p* = 0.097). The receiver operating characteristic (ROC) curve analysis identified optimal cutoff values of ≤67.2 HU for axial measurements (sensitivity 77.9%, specificity 77.8%) and ≤59.7 HU for coronal measurements (sensitivity 75.3%, specificity 80.0%) through Youden index maximization.

## 4. Discussion

We have developed a coronal CT attenuation measurement method for osteoporosis screening at the proximal femur. Bland–Altman analysis revealed a limit of agreement (CR = 9.09 HU) between this method and the established axial measurement, indicating that the two methods are not interchangeable for providing identical absolute HU values. Nonetheless, the strong correlation (r = 0.919) between them confirms that our proposed method captures the same trend in bone density. Crucially, both methods demonstrated comparable correlations with proximal femoral BMD T-scores (0.766 [95% CI: 0.730–0.808] vs. 0.736 [95% CI: 0.697–0.771], *p* = 0.195). Most importantly for clinical application, both techniques exhibited good predictive performance for osteoporosis diagnosis, with no statistically significant difference in AUC values between the two methods (0.864 vs. 0.850, *p* = 0.097).

In recent years, there has been a growing body of research utilizing routine clinical CT examinations for osteoporosis screening, capitalizing on their inherent advantages of avoiding additional radiation exposure and superior clinical accessibility [23]. Compared to lumbar attenuation measurements, proximal femur measurements can directly reflect regional bone density and demonstrate a strong correlation with hip fracture incidence, which is one of the leading causes of mortality in elderly populations [24,25]. Currently, CT attenuation measurements of the proximal femur predominantly employ axial plane acquisition. The axial measurement method described by Christensen et al. [17] requires averaging values across multiple axial planes. While our previous study demonstrated that a single slice can adequately reflect BMD correlation [8], the inherent complexity of multi-plane axial measurements may introduce interpretation bias when performed by radiologists (as opposed to orthopedic specialists). Furthermore, single axial measurements impose certain limitations for osteoporosis screening, whereas multi-planar (including coronal, sagittal plane) CT attenuation value assessments may provide a more comprehensive reflection of regional bone density status. This has been validated in vertebral CT attenuation value measurements [26]. Therefore, there is a need to develop a simplified alternative to conventional measurements for reliable bone quality assessment. The coronal plane measurement technique offers inherent advantages for femoral neck assessment due to the anatomical orientation of the proximal femur. Given that the proximal femur width in the sagittal plane is smaller than its width in the coronal plane [27,28], coronal imaging requires fewer slices to capture the plane containing maximal cancellous bone area compared to sagittal imaging. This anatomical characteristic may reduce the operator-dependent variability in selecting the optimal measurement plane.

Our coronal plane measurement method demonstrated correlation coefficients with T-score and screening AUC values that were comparable to those reported in the previous literature. A meta-analysis of 41 studies revealed significantly higher accuracy for proximal femur CT scans compared to other anatomical sites in osteoporosis screening [25]. Among the 8 studies investigating proximal femoral CT attenuation values for osteoporosis screening, the average correlation coefficient with BMD was 0.70 (95% CI: 0.57–0.82), and the mean AUC was 0.79 (95% CI: 0.72–0.87). In our study, the corresponding correlation coefficient and AUC values of coronal measurement were 0.736 and 0.850. Moreover, while maintaining equivalent diagnostic efficacy, our coronal plane measurement method offers potential operational simplicity.

In our analysis, the coronal measurement method demonstrated comparable BMD assessment performance to the axial measurement method without statistically significant differences (*p* > 0.05). Despite demonstrating comparable diagnostic utility, the two methods showed a clinically relevant discrepancy in absolute values, with a coefficient of repeatability of 9.09 HU (95% CI: 8.63–9.61). This phenomenon may be associated with the distinct distribution patterns and density variations in trabecular bone between the femoral neck and intertrochanteric regions. During coronal plane measurements, the relatively lower-density Ward’s triangle area is more likely to be included in the region of interest, resulting in an overall reduction in attenuation values [29,30]. The inclusion of this osteopenic region disproportionately dilutes the average attenuation value, resulting in an overall reduction in attenuation values. Based on the Youden index calculation, the optimal thresholds for osteoporosis screening were determined to be ≤59.7 HU for coronal measurements and ≤67.2 HU for axial measurements (consistent with our team’s prior findings). This demonstrates that after appropriate calibration, both methods can achieve effective screening performance through their respective diagnostic thresholds. The axial optimal threshold in this study was highly consistent with our previous study [8], which made our results more convincing. However, different scanning equipment and parameters can affect this diagnostic threshold, which may require pre-calibration before application. For instance, this study employed anisotropic 5 mm slices, which may introduce partial volume effects. This phenomenon could influence measurement precision, particularly in regions with complex trabecular architecture. If thinner slice parameters were used, the diagnostic thresholds might require corresponding adjustments.

Currently, several methods exist for equivalent BMD measurement of the proximal femur. In addition to the conventional axial measurement method previously employed, Anderson et al. [19] also explored a coronal measurement approach at the proximal femur in a retrospective study of 240 patients, reporting high sensitivity (0.92) and specificity (0.93) for osteoporosis prediction using their method. However, their measurement was based on the QCT, which has relatively limited clinical adoption. Moreover, they utilized an irregular ROI of the proximal femur, which may pose a challenge in ROI selection. Park et al. [20] developed a three-dimensional analysis method for assessing proximal femoral BMD, offering a more comprehensive evaluation of regional BMD compared to single-plane measurements. However, this approach still requires specialized software, limiting its clinical utility for routine practice. As demonstrated by Keisuke et al. [18,31], who developed a deep learning-based system for equivalent BMD measurement at the proximal femur and osteoporosis diagnosis using CT images, their method shows high clinical efficacy. However, their method may require image format conversion and preprocessing, which increases operational complexity. The goal of osteoporosis screening is to efficiently identify high-risk individuals using widely accessible and low-cost examinations. In this context, the coronal measurement method proposed in our study, based on routine CT scans, is simpler and more practical, making it potentially more suitable as an initial screening tool.

This study has several limitations. First, the retrospective design may introduce selection bias, as evidenced by the gender imbalance (76.8% female), although this distribution reflects the known epidemiology of osteoporosis. Additionally, the vast majority of patients had either osteopenia or osteoporosis. This selection bias indeed limits the generalizability of our findings to broader populations. Furthermore, this study did not control for potential confounding factors such as comorbidities, which may have influenced the interpretation of the results. Second, this study did not evaluate the correlation between proximal femoral CT attenuation values and hip fracture risk; therefore, establishing this relationship remains an objective for future investigation. Third, although coronal and axial CT measurements showed comparable diagnostic accuracy, further validation in a prospective study with a pre-defined non-inferiority margin is still needed. Finally, the next step involves integrating traditional manual measurement techniques with AI technology, streamlining the process, and embedding it into daily medical practice to ensure convenient and efficient use in routine screenings.

## 5. Conclusions

In summary, our findings demonstrate that the coronal CT attenuation measurement method for the proximal femur represents a viable approach for opportunistic osteoporosis screening. This method demonstrated no statistically significant difference from the conventional axial measurement in either its correlation with femoral BMD T-scores or its diagnostic accuracy for osteoporosis. In the context of widespread routine CT examinations, this method can be seamlessly integrated into radiologists’ standardized workflow, potentially improving osteoporosis detection rates and facilitating subsequent clinical evaluation and intervention.

## Figures and Tables

**Figure 1 diagnostics-15-02794-f001:**
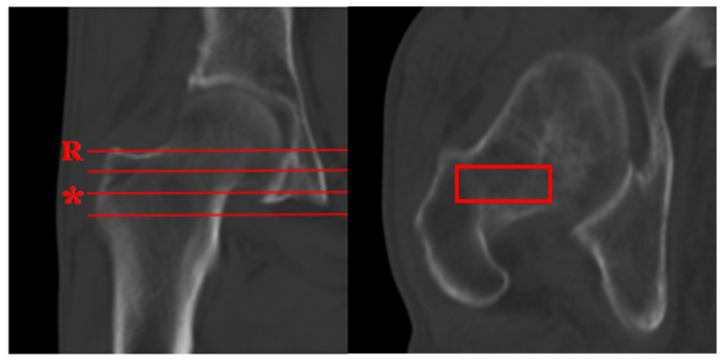
The axial measurement plane (*) was defined as the second axial slice located 10 mm inferior to the reference slice (R), where the femoral neck cortical bone first appears. A red rectangular ROI was positioned on the axial measurement plane (*) to encompass the maximal cancellous bone area while excluding cortical bone, and the mean CT attenuation value was recorded.

**Figure 2 diagnostics-15-02794-f002:**
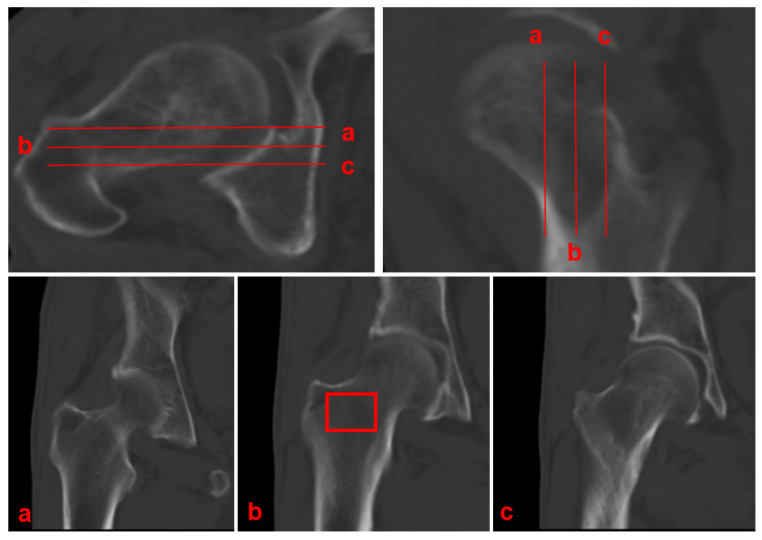
The coronal measurement plane should be selected at the level where the femoral neck and intertrochanteric region contain the maximum amount of cancellous bone within the red rectangular ROI (**b**). As shown in (**b**) (compared to (**a**,**c**)), this plane clearly demonstrates more abundant cancellous bone and should therefore serve as the optimal measurement plane.

**Figure 3 diagnostics-15-02794-f003:**
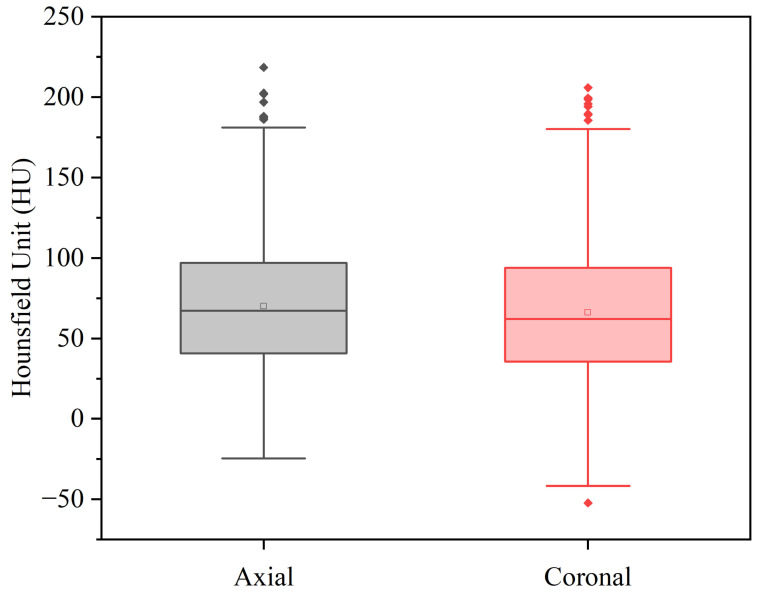
Median values were 67.20 HU (interquartile range [IQR]: 40.80–97.00) for axial measurements and 62.05 HU (IQR: 35.50–93.90) for coronal measurements.

**Figure 4 diagnostics-15-02794-f004:**
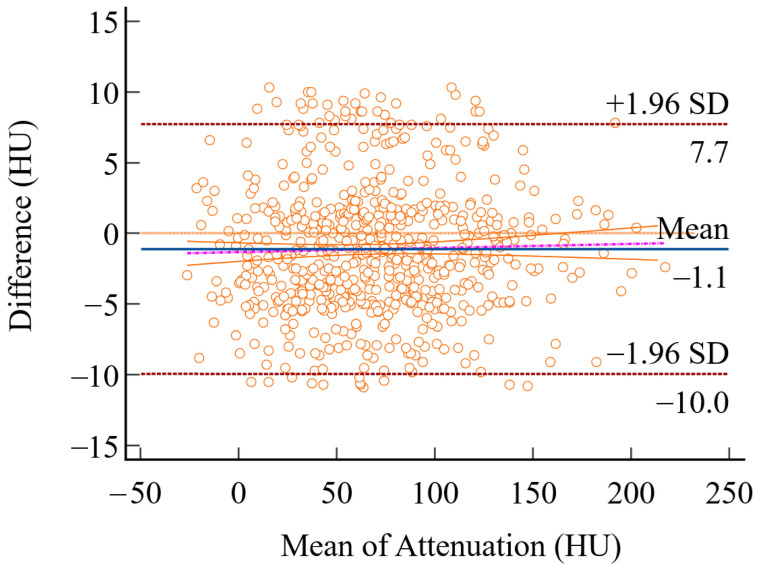
Bland-Altman plot assessing the agreement between axial and coronal measurements. The x-axis represents the mean of both measurements, and the y-axis shows the difference between them (axial minus coronal). Each granular circle represents a single paired measurement. The analysis showed a Coefficient of Repeatability (CR) = 9.09 (95% CI: 8.63 to 9.61).

**Figure 5 diagnostics-15-02794-f005:**
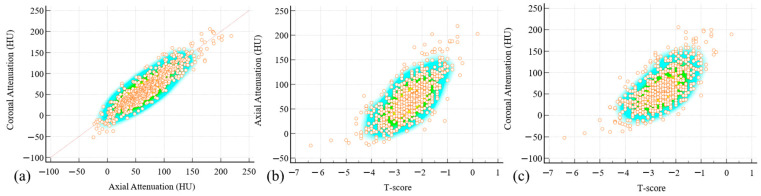
Pearson’s coefficient (r) between axial and coronal attenuation was 0.919 (95% CI: 0.907–0.930, *p* < 0.001) (**a**). The correlation coefficients between CT attenuation values (axial (**b**)/coronal (**c**)) and T-scores were 0.766 (95% CI: 0.730–0.808) and 0.736 (95% CI: 0.697–0.771). Each granular circle represents an individual data point. The shaded background color intensity reflects the density distribution of the data points, with darker areas indicating a higher local density.

**Figure 6 diagnostics-15-02794-f006:**
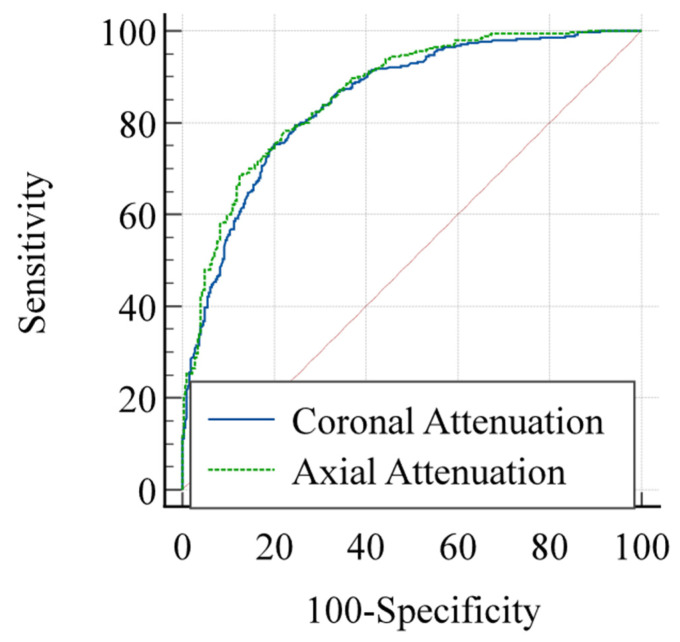
The axial measurements yielded an area under the curve (AUC) of 0.864 (95% CI: 0.834–0.891) for predicting osteoporosis, while the coronal measurements achieved an AUC of 0.850 (95% CI: 0.803–0.878). The diagonal red line represents the reference line of no discriminatory ability (AUC = 0.5).

**Table 1 diagnostics-15-02794-t001:** Patient demographics and the comparison among the normal BMD, osteopenia, and osteoporosis BMD, bone mineral density; HU, Hounsfield unit. * *p* < 0.05 means a significant difference among these groups.

Measurement	ICC	95% CI	*p **
Axial attenuation	0.843	0.810–0.871	<0.001
Coronal attenuation	0.862	0.833–0.887	<0.001

ICC, intraclass correlation coefficient; CI, confidence interval.

**Table 2 diagnostics-15-02794-t002:** Inter-observer reliability for each measurement.

Patient Information	Overall (*n* = 708)	Normal BMD (a) (*n* = 14)	Osteopenia (b) (*n* = 108)	Osteoporosis (c) (*n* = 592)	*p* *	*p* _a−b_	*p* _a−c_	*p* _b−c_
Gender (M/F)	164/543	2/12	26/75	136/456	0.605	0.342	0.447	0.543
Interval time (days)	20 (3, 83)	35 (1, 83)	38 (6, 93)	19 (3, 78)	0.146	0.458	0.998	0.050
Age (years)	64.24 ± 11.29	51.75 ± 11.82	59.60 ± 11.75	65.62 ± 10.98	0.007	0.204	0.017	0.027
Axial attenuation (HU)	70.67 ± 42.30	170.02 ± 30.81	99.73 ± 36.94	63.37 ± 38.13	<0.001	<0.001	<0.001	<0.001
Coronal attenuation (HU)	66.13 ± 44.83	167.78 ± 44.69	96.26 ± 38.30	60.88 ± 43.80	<0.001	<0.001	<0.001	<0.001
Femoral neck T-score	−2.52 ± 0.83	−0.55 ± 0.30	−1.81 ± 0.44	−2.68 ± 0.76	<0.001	<0.001	<0.001	<0.001

BMD, bone mineral density; HU, Hounsfield unit. * *p* < 0.05 means a significant difference among these groups. For pairwise comparisons following a significant ANOVA (LSD test) or Kruskal-Wallis test (Dunn’s test), the *p*-values are denoted as follows: *p*_a−b_ for normal BMD vs. osteopenia; *p*_b−c_ for osteopenia vs. osteoporosis; *p*_a−c_ for normal BMD vs. osteoporosis.

## Data Availability

The data presented in this study are available on request from the corresponding author due to ongoing research and the data forming part of an ongoing study.

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
