# Peer review of "Coronal CT Attenuation Measurement for Osteoporosis Screening at the Proximal Femur: A Comparative Study with the Axial Approach"

_diagnostics, 2025, doi:10.3390/diagnostics15212794_

Round 1
Reviewer 1 Report
Comments and Suggestions for Authors
Review diagnostics-3943717-peer-review-v1
The aim of the paper “Coronal CT Attenuation Measurement for Osteoporosis Screening at the Proximal Femur: A Comparative Study with the Axial Approach” was to evaluate a coronal CT attenuation measurement method for osteoporosis screening at the proximal femur and compare its performance with the conventional axial approach. The study analyzed 708 patients who underwent both CT and DXA within six months and found that coronal CT measurements correlated strongly with axial measurements and showed comparable diagnostic performance for identifying osteoporosis. This study presents an interesting and clinically meaningful investigation, supported by a comprehensive methodological framework and robust quantitative analysis.
The figures and tables are well designed, providing clear visualizations of the measurement planes (Figures 1–2), correlation and diagnostic plots (Figures 3–6), and systematically organized data. The references are current (2023–2025) and include relevant prior studies in CT-based osteoporosis research. No excessive self-citation was observed. I have only few minor comments.
Abstract.
Line 28 – describe what HU is?
Introduction. Please, use smoother transitions between DXA, QCT, and CT-based screening. Moreover, please clarify the rationale for using coronal measurements.
Methods. The gender distribution (76.6% female) and the predominance of osteopenia/ osteoporosis cases limit external validity to a broader population.
The paper lacks sufficient detail on scanner calibration and standardization, which affects reproducibility across institutions.
Results are clear and present, and I have no points for improvement.
Discussion. Please discuss the clinical implications of small HU differences in more depth. Reduce repetition in discussion (lines 235–262 repeat several results).
Author Response
The aim of the paper “Coronal CT Attenuation Measurement for Osteoporosis Screening at the Proximal Femur: A Comparative Study with the Axial Approach” was to evaluate a coronal CT attenuation measurement method for osteoporosis screening at the proximal femur and compare its performance with the conventional axial approach. The study analyzed 708 patients who underwent both CT and DXA within six months and found that coronal CT measurements correlated strongly with axial measurements and showed comparable diagnostic performance for identifying osteoporosis. This study presents an interesting and clinically meaningful investigation, supported by a comprehensive methodological framework and robust quantitative analysis.
The figures and tables are well designed, providing clear visualizations of the measurement planes (Figures 1–2), correlation and diagnostic plots (Figures 3–6), and systematically organized data. The references are current (2023–2025) and include relevant prior studies in CT-based osteoporosis research. No excessive self-citation was observed. I have only few minor comments.
Response: We sincerely thank you for your positive assessment of our manuscript. Below are our point-by-point responses to your comments and the corresponding revisions we have made.
Abstract.
1. Line 28 – describe what HU is?
Response: Thank you for your suggestion. The full term "Hounsfield Units (HU)" has been added upon its first appearance in the text (Line 28) to ensure clarity for all readers.
Introduction.
2. Please, use smoother transitions between DXA, QCT, and CT-based screening. Moreover, please clarify the rationale for using coronal measurements.
Response: Thank you for your suggestions. We have revised the text to improve the logical flow from DXA to QCT and then to CT-based screening (Lines 48-56). Additionally, we have clarified the rationale for using coronal measurements as follows: 1) Measurements in other planes can complement axial measurements; 2) The coronal plane of the proximal femur covers a broader anatomical area, enabling a more comprehensive assessment of overall bone mass (Lines 64-70).
Methods.
3. The gender distribution (76.6% female) and the predominance of osteopenia/ osteoporosis cases limit external validity to a broader population.
Response: Thank you for the suggestion. We fully agree with this observation. As this is a retrospective study, the enrolled participants were exclusively patients clinically suspected of having osteoporosis and referred for DXA examination. Given that osteoporosis predominantly affects postmenopausal women, our cohort consequently has a higher proportion of females and is concentrated in individuals with abnormal bone mass. This selection bias indeed limits the generalizability of our findings to broader populations, particularly males and individuals with normal bone mass. We have explicitly acknowledged this limitation in the Discussion section and recommend that future prospective studies be conducted in more representative populations for further validation (Lines 298-304).
4. The paper lacks sufficient detail on scanner calibration and standardization, which affects reproducibility across institutions.
Response: Thank you for your suggestion. We have expanded the Methods section to include detailed CT scanning parameters (Dual Source Computed Tomography DEFINITION, tube voltage 120 kV, a slice thickness of 5 mm, Siemens) (Lines 106-109). Additionally, we have acknowledged in the Discussion that variations in scanning protocols across different CT systems may affect the generalizability of diagnostic thresholds (Lines 273-278).
Discussion.
5. Please discuss the clinical implications of small HU differences in more depth. Reduce repetition in discussion (lines 235–262 repeat several results).
Response: Thank you for your suggestions. We have provided additional explanation from an anatomical perspective regarding the differences in ROI placement between axial and coronal measurements (Lines 259-267). Furthermore, we have removed the repetitive sections in the Discussion (Lines 235-262).
Reviewer 2 Report
Comments and Suggestions for Authors
This study compares coronal and axial CT attenuation measurements at the proximal femur for opportunistic osteoporosis screening against DXA-based T-scores. The proposed coronal method aims to simplify workflow and expand screening potential in clinical settings.
- Incremental novelty — builds upon prior axial HU studies rather than introducing new imaging or analytical methodology.
- The coronal method’s clinical advantage remains theoretical, as diagnostic equivalence (AUC 0.850 vs. 0.864) does not clearly justify a paradigm shift.
- Majority (76.6%) female and overrepresentation of osteopenia/osteoporosis
- No control for confounders such as comorbidities
- DXA and CT within 6 months — bone density could shift, particularly in elderly or osteoporotic cohorts.
- Clinical relevance of CR = 9.09 HU not discussed
- Correlation vs. agreement: strong Pearson’s r (0.919) does not prove interchangeability
- AUC difference (0.864 vs. 0.850, p = 0.097) interpreted as equivalence — but non-inferiority margin not defined.
- The claim of “superior operational simplicity” is qualitative; no survey or workflow timing data support it
- Should better discuss standardization needs (ROI size, reconstruction slice thickness, HU calibration) for reproducibility across centers.
Author Response
1. Incremental novelty — builds upon prior axial HU studies rather than introducing new imaging or analytical methodology.
Response: Thank you for your suggestion. We agree that our study builds upon the established foundation of axial HU measurements. While prior studies have primarily relied on axial or complex 3D coronal measurements, we specifically designed and validated a simplified coronal approach that establishes a new clinical pathway for opportunistic screening. This pathway offers two distinct advantages: 1) It provides an operationally simpler alternative for routine clinical settings where time and software resources are limited; 2) It serves as a complementary method to existing approaches, offering clinicians flexibility to choose the most appropriate measurement plane based on image quality and anatomical presentation. We believe this method also demonstrates reasonably good diagnostic performance and could potentially create new possibilities for clinical implementation.
2. The coronal method’s clinical advantage remains theoretical, as diagnostic equivalence (AUC 0.850 vs. 0.864) does not clearly justify a paradigm shift.
Response: Thank you for your suggestion. We acknowledge that the claim of operational simplicity requires further objective validation. The clinical advantage of the coronal method is primarily inferred from its simpler workflow—aligning the ROI with the long axis of the femoral neck on a single coronal image is intuitively easier and potentially faster than navigating multiple axial slices. While the diagnostic equivalence (AUC) alone may not justify a paradigm shift, the combination of comparable accuracy and significantly reduced operational complexity presents a compelling case for its utility. Furthermore, in future clinical practice, a complementary approach combining both axial and coronal measurements could be employed for cases with diagnostic uncertainty to enhance screening robustness.
3. Majority (76.6%) female and overrepresentation of osteopenia/osteoporosis
Response: Thank you for the suggestion. We fully agree with this observation. As this is a retrospective study, the enrolled participants were exclusively patients clinically suspected of having osteoporosis and referred for DXA examination. Given that osteoporosis predominantly affects postmenopausal women, our cohort consequently has a higher proportion of females and is concentrated in individuals with abnormal bone mass. This selection bias indeed limits the generalizability of our findings to broader populations, particularly males and individuals with normal bone mass. We have explicitly acknowledged this limitation in the Discussion section and recommend that future prospective studies be conducted in more representative populations for further validation (Lines 298-304).
4. No control for confounders such as comorbidities.
Response: Thank you for your suggestion. Indeed, comorbidities and medications that affect bone metabolism were not controlled for in this retrospective analysis. We will add this as a study limitation in the discussion (Lines 302-304) and suggest that the influence of such confounders be rigorously investigated in future, larger-scale prospective studies.
5. DXA and CT within 6 months — bone density could shift, particularly in elderly or osteoporotic cohorts.
Response: Thank you for your comment. The median (interquartile range) time interval between DXA and CT examinations was 20 (3, 83) days across all participants (Table 2). Bone density is unlikely to change significantly within this period. However, we acknowledge that for a subset of patients with longer intervals between scans, bone density changes might have occurred, which represents a potential limitation of this study.
6. Clinical relevance of CR = 9.09 HU not discussed
Response: Thank you for your suggestion. We have expanded the discussion on the CR of 9.09 HU (Lines 259-267). While this indicates the two methods are not interchangeable for absolute HU measurement, their clinical utility is preserved by employing method-specific thresholds (≤59.7 HU for coronal, ≤67.2 HU for axial). The establishment of these distinct, validated cut-off values effectively calibrates for the systematic difference, allowing each method to be used independently for effective screening.
7. Correlation vs. agreement: strong Pearson’s r (0.919) does not prove interchangeability
Response: Thank you for your suggestion. We fully agree that a strong correlation does not equate to agreement or interchangeability. The high Pearson's r value primarily indicates that the two methods move in the same direction, but as the Bland-Altman analysis (CR=9.09 HU) shows, their absolute values are not interchangeable. Our aim was to demonstrate the comparable diagnostic performance of the two methods, and the text has been clarified to avoid any misinterpretation (Lines 214-223).
8. AUC difference (0.864 vs. 0.850, p = 0.097) interpreted as equivalence — but non-inferiority margin not defined.
Response: Thank you for the suggestion. We fully understand that the lack of statistical significance (p > 0.05) does not directly demonstrate clinical equivalence. In the revised manuscript, we have revised the wording from "equivalence" to "no statistically significant difference" (Lines 214-223) and have explicitly stated that this finding requires further validation in a prospective study with a pre-defined non-inferiority margin (Lines 307-308). We concur that formally defining a non-inferiority margin warrants dedicated investigation in future studies.
9. The claim of “superior operational simplicity” is qualitative; no survey or workflow timing data support it
Response: Thank you for your suggestion. The wording has been revised from "superior operational simplicity" to "potential operational advantage" in the manuscript (Lines 255-256). We acknowledge that the actual clinical benefit requires future validation through formal user experience studies and quantitative workflow timing analyses.
10. Should better discuss standardization needs (ROI size, reconstruction slice thickness, HU calibration) for reproducibility across centers.
Response: Thank you for your suggestion. We agree that different scanning equipment and parameters may potentially influence HU measurements. We have addressed this point in the Discussion section, emphasizing that the thresholds proposed in our study are specific to our particular scanning protocol (Lines 273-276). Furthermore, we have now provided more comprehensive details regarding the scanning equipment and parameters in the Methods section for reference (Lines 106-109).
Reviewer 3 Report
Comments and Suggestions for Authors
Coronal CT Attenuation Measurement for Osteoporosis Screening at the Proximal Femur: A Comparative Study with the Axial Approach
1) The presented work is a single-center, retrospective diagnostic accuracy study of 708 patients who received both proximal femur computed tomography (CT) and dual-energy X-ray absorptiometry (DXA) scans in 6 months. The primary goal was to evaluate a novel coronal Hounsfield Unit (HU) measurement method. The authors compared the index test, axial measurements and T-scores in terms of reliability, agreement, and diagnostic performance for osteoporosis screening.
The manuscript does not explicitly state a null hypothesis. Perhaps, “There is no significant difference between coronal and axial CT attenuation measurements in terms of (a) correlation with femoral T-scores and (b) diagnostic accuracy (AUC) for identifying osteoporosis” would be appropriate? Consider framing this as a non-inferiority hypothesis with a pre-specified margin to strengthen the claim of interchangeability.
2) The study demonstrates that a simple coronal CT attenuation measurement at the proximal femur is a reliable and effective method for opportunistic osteoporosis screening. This novel method is nearly identical to the established axial technique in diagnostic performance (r ≈ 0.92) and no statistically significant difference in diagnostic accuracy (AUC 0.850 vs. 0.864). The authors provide HU cutoffs (≤59.7 HU for coronal, ≤67.2 HU for axial) balancing sensitivity and specificity for screening.
3) Opportunistic screening on routine CT scans is a well-established methodology to increase diagnostic yield without additional cost or radiation exposure. While lumbar spine (L1) attenuation is a widely used screening target, the presented study reinforces the utility of proximal femur measurement, which has two key advantages:
- Hip attenuation directly reflects bone quality at a site of fragility fractures, and is directly relevant for orthopedic surgical planning.
- The method is applicable on pelvic or hip CT scans, where the lumbar spine is not in the field of view, and it avoids confounders of the aging spine.
The authors expand existing literature by providing a simplified, accessible method.
- Christensen et al. (10.1097/CORR.0000000000000480) studied the correlation between proximal femur HU and DXA, offering a coronal alternative.
- Compared to the 3D or specialized software approaches described by Park et al. (10.1371/journal.pone.0262025), the proposed 2D coronal ROI method requires can be implemented on any standard PACS, lowering the barrier to clinical adoption.
4) Suggestions to further refine the article are as follows:
- HU values are highly dependent on scanner parameters (kVp, reconstruction kernel, iterative reconstruction level) and manufacturer. The study used a single scanner type at 120 kVp. Consider discussing the potential need for protocol-specific or calibrated thresholds for broader applicability. Discuss the influence of anisotropic (5 mm) slices on measurement accuracy.
- Provide 95% confidence intervals for the AUC values to better illustrate their overlap. Consider including a calibration analysis or decision curve analysis to assess the clinical utility of the proposed thresholds.
- In the Discussion, the correlation between axial and coronal CT attenuation measurement methods (r = 0.919) is described as "moderate." This is typically considered a very strong or excellent correlation and should be described consistently throughout the manuscript.
- The description of ROI placement could be enhanced by providing a brief, step-by-step protocol.
5) The conclusion that coronal CT attenuation measurement is a “viable approach for opportunistic osteoporosis screening” is consistent with the evidence provided.
6) The figures and tables effectively communicate the key results. A minor improvement would be to add the 95% CI for the AUCs.
7) The study appropriately obtained IRB approval with a waiver of informed consent for its retrospective design. The data availability statement is standard. To facilitate meta-analyses, the authors could consider making the de-identified measurement data available in a public repository.
Author Response
1. The manuscript does not explicitly state a null hypothesis. Perhaps, “There is no significant difference between coronal and axial CT attenuation measurements in terms of (a) correlation with femoral T-scores and (b) diagnostic accuracy (AUC) for identifying osteoporosis” would be appropriate? Consider framing this as a non-inferiority hypothesis with a pre-specified margin to strengthen the claim of interchangeability.
Response: Thank you for the suggestion. Regarding the null hypothesis, we have explicitly stated it in the revised manuscript as follows: "There is no significant difference in the diagnostic performance for osteoporosis between the coronal and axial CT attenuation measurement methods.” (Lines 75-76). Additionally, the Conclusion section has been revised to align with the null hypothesis (Lines 316-318). We fully agree that a formal non-inferiority design with a pre-specified margin would provide the strongest evidence for interchangeability. However, defining a clinically justified non-inferiority margin for CT attenuation values a priori was challenging for this initial validation study, as such a margin has not been established in the literature. Therefore, the primary aim of our work was to provide preliminary evidence of comparable diagnostic performance and to establish the foundation for future studies (Lines 306-308).
2. HU values are highly dependent on scanner parameters (kVp, reconstruction kernel, iterative reconstruction level) and manufacturer. The study used a single scanner type at 120 kVp. Consider discussing the potential need for protocol-specific or calibrated thresholds for broader applicability. Discuss the influence of anisotropic (5 mm) slices on measurement accuracy.
Response: Thank you for the suggestion. We fully agree that different scanning equipment and parameters may potentially influence HU measurements. We have addressed this point in the Discussion section, emphasizing that the thresholds proposed in our study are specific to our particular scanning protocol (Lines 273-275). Regarding the impact of anisotropic 5 mm slice thickness, we have added new discussion content (Lines 275-278) acknowledging its potential effect on measurement accuracy through partial volume effects.
3. Provide 95% confidence intervals for the AUC values to better illustrate their overlap. Consider including a calibration analysis or decision curve analysis to assess the clinical utility of the proposed thresholds.
Response: We fully agree with your suggestion. The 95% confidence intervals for the AUC values have now been added to the Results section (Lines 202-203).
4. In the Discussion, the correlation between axial and coronal CT attenuation measurement methods (r = 0.919) is described as "moderate." This is typically considered a very strong or excellent correlation and should be described consistently throughout the manuscript.
Response: Thank you for this insightful suggestion. We agree with your assessment and have revised the description of the correlation from "moderate" to "strong" throughout the manuscript to accurately reflect the strength of the association (r = 0.919) between the axial and coronal CT attenuation measurement methods (Line 217).
5. The description of ROI placement could be enhanced by providing a brief, step-by-step protocol.
Response: Thank you for the suggestion. A concise step-by-step protocol for ROI placement has been added to the Methods section to improve reproducibility and clarity (Lines 120-130).
6. The figures and tables effectively communicate the key results. A minor improvement would be to add the 95% CI for the AUCs.
Response: Thank you for your suggestion. The 95% confidence intervals for the AUC values have now been added to the Results section.
7. The study appropriately obtained IRB approval with a waiver of informed consent for its retrospective design. The data availability statement is standard. To facilitate meta-analyses, the authors could consider making the de-identified measurement data available in a public repository.
Response: Thank you for your suggestion. In accordance with the journal's policy and to promote transparency, we will consider depositing the de-identified measurement data underlying the findings of this study in a public repository.
Round 2
Reviewer 2 Report
Comments and Suggestions for Authors
The manuscript is suitable for publication in its current form.
Reviewer 3 Report
Comments and Suggestions for Authors
The authors have addressed the concerns raised in my initial review.
The explicit framing of the comparative aims, the addition of 95% confidence intervals for all AUC values strengthens the statistical reporting. The ROI placement protocol (line 121) enhances reproducibility. The revised Discussion appropriately contextualizes the findings, acknowledging scanner-parameter dependence and potential partial volume effects from anisotropic slices.
While calibration and decision curve analyses were not incorporated, the rationale for their omission is reasonable. The data availability statement remains unchanged, but acceptable given journal policies.
The manuscript now presents a methodologically sound, clinically relevant contribution to opportunistic osteoporosis screening methodology. The simplified coronal approach offers practical advantages and is supported by the evidence presented.